# Therapeutic Advances in Psoriasis: From Biologics to Emerging Oral Small Molecules

**DOI:** 10.3390/antib13030076

**Published:** 2024-09-14

**Authors:** Francesco Ferrara, Chiara Verduci, Emanuela Laconi, Andrea Mangione, Chiara Dondi, Marta Del Vecchio, Veronica Carlevatti, Andrea Zovi, Maurizio Capuozzo, Roberto Langella

**Affiliations:** 1Pharmaceutical Department, Asl Napoli 3 Sud, Dell’amicizia Street 72, 80035 Nola, Italy; m.capuozzo@aslnapoli3sud.it; 2IRCCS Humanitas Research Hospital, Manzoni Street 56, 20089 Rozzano, Italy; verducichiara92@gmail.com; 3Pharmaceutical Department, ASST Nord Milano, E. Bassini Hospital, Massimo Gorki Street 50, 20092 Cinisello Balsamo, Italy; emanuela.laconi@asst-nordmilano.it; 4Pharmaceutical Department, ASST Valle Olona, Busto Arsizio Hospital, Arnaldo da Brescia 1 Street, 21052 Busto Arsizio, Italy; andrea.mangione.1@gmail.com; 5Pharmaceutical Department, ASST Ovest Milanese, Legnano Hospital, Papa Giovanni Paolo II Street, 20025 Legnano, Italy; chiara.dondi95@gmail.com (C.D.); delvecchiomarta@gmail.com (M.D.V.); 6Hospital Pharmacy Department, ASST Fatebenefratelli-Sacco, V. Buzzi Hospital, Castelvetro Street 28, 20154 Milano, Italy; veronica.carlevatti@asst-fbf-sacco.it; 7Ministry of Health, Viale Giorgio Ribotta 5, 00144 Rome, Italy; zovi.andrea@gmail.com; 8Italian Society of Hospital Pharmacy (SIFO), SIFO Secretariat of the Lombardy Region, Via Carlo Farini 81, 20159 Milan, Italy; roberto.langella87@gmail.com

**Keywords:** psoriasis, inhibitors, plaque psoriasis, JAK/STAT, IL-17, IL-23

## Abstract

Psoriasis is a persistent, inflammatory condition affecting millions globally, marked by excessive keratinocyte proliferation, immune cell infiltration, and widespread inflammation. Over the years, therapeutic approaches have developed significantly, shifting from conventional topical treatments and phototherapy to more sophisticated systemic interventions such as biologics and, recently, oral small-molecule drugs. This review seeks to present a comprehensive investigation of the existing psoriasis treatment options, focusing on biologic agents, oral small molecules, and emerging treatments. Several categories of biologic treatments have received regulatory approval for psoriasis, including TNF-α, IL-17, IL-12/23, and IL-23 inhibitors. Biologics have revolutionized the treatment of psoriasis. These targeted therapies offer significant improvement in disease control and quality of life, with acceptable safety profiles. However, limitations such as cost, potential immunogenicity, and administration challenges have driven the exploration of alternative treatment modalities. Oral small molecules, particularly inhibitors of Janus kinase (JAK), have emerged as options due to their convenience and efficacy. These agents represent a paradigm shift in the management of the condition, offering oral administration and targeted action on specific signaling pathways. In addition to existing therapies, the review explores emerging treatments that hold promise for the future of psoriasis care. These include innovative small-molecule inhibitors. Early-stage clinical trials suggest these agents may enhance outcomes for psoriasis patients. In conclusion, the therapeutic landscape of psoriasis is rapidly evolving, emphasizing targeted, patient-centered treatments. Ongoing research and development are expected to lead to more personalized and effective management strategies for this complex condition.

## 1. Introduction

Psoriasis, a multifactorial and chronic skin disease mediated by biological defense mechanisms, commonly affects 3% of the general population [1]. It is linked with significant comorbidities, productivity loss, work limitations, and healthcare costs exceeding USD 40 billion annually [2]. Many studies in the literature have indicated that the prevalence of psoriasis ranges from approximately 30 cases per 100,000 inhabitants to about 320 cases per 100,000 inhabitants. This condition demonstrates correlations with factors such as age, gender, and ethnic background, as well as a variety of genetic and environmental influences [3,4,5]. This persistent negative impact on quality of life and financial burden underscores the necessity for effective long-term disease management. Disease severity is influenced by factors such as lesion extent, localization, and associated conditions like psoriatic arthritis [6,7]. Successful treatment involves achieving either remission or near remission, correlating with enhanced quality of life. Moreover, an increasing body of research has suggested that psoriasis represents a systemic condition with various accompanying illnesses, including cardiovascular ailments, psoriatic arthritis, metabolic syndrome, depression, and anxiety [8,9]. Hence, it is imperative to address both the comorbidities associated with psoriasis and the disease itself in order to devise timely intervention and treatment approaches. Recent years have seen the development of highly effective targeted therapies, including conventional, biological, and oral small-molecule treatments. These biological therapies encompass various classes, such as tumor necrosis factor (TNF)-α inhibitors (e.g., etanercept, adalimumab, golimumab, certolizumab pegol, and infliximab), Interleukin (IL)-17 inhibitors (e.g., brodalumab, ixekizumab, and secukinumab), IL-23 inhibitors (e.g., Risankizumab, tildrakizumab, and guselkumab), and IL-12/23 inhibitor (ustekinumab) [10,11,12]. While these innovations have elevated treatment efficacy and safety standards, there remains a demand for novel therapies enabling superior skin remission and durability. Furthermore, many treatments experience declining efficacy over time, necessitating the exploration of alternative options. Recent advances in understanding psoriasis pathophysiology have spurred the development of additional therapeutic targets, including IL-36 inhibitors, phosphodiesterase (PDE)-4 inhibitors, Janus kinase (JAK) inhibitors, tyrosine kinase 2 (TYK2) inhibitors, RORγt inhibitors, and A3 adenosine receptor agonists [13,14,15]. This narrative review explores the latest progress in biologicals, oral small molecules, and new biosimilar drugs for psoriasis treatment.

This review aims to discuss all treatments for moderate to severe psoriasis, starting with biological drugs and monoclonal antibodies and extending to the most recent oral treatments with small molecules. The metabolic aspect is discussed briefly to introduce the pathology and then to address the actual pharmacological treatment. Obsolete treatments, such as rarely used oral medications for moderate to severe forms, as well as topical treatments with creams that have no systemic curative role but only symptomatic relief, are not considered. To date, the efficacy and effectiveness of each drug can vary based on the patient’s clinical conditions, making it impossible to evaluate or prefer one drug over another universally. The patient’s clinical conditions and those related to therapeutic appropriateness, along with associated costs, must always be considered by the prescribing physician. For example, in treatment-naive patients, etanercept or adalimumab, which now also have biosimilar drugs available, should always be used as first-line treatments when efficacy is equal, compared to newer, higher-cost drugs.

## 2. Currently Available Biologic Therapy for Psoriasis

The outcomes derived from numerous genome-wide association studies (GWAS) and clinical trials corroborate the pivotal involvement of TNF/IL-17/IL-23 signaling pathways in the psoriasis etiology [16,17,18,19]. TNF-α serves as a crucial inflammatory cytokine highly evolved in psoriatic lesions, playing a central role in the development of psoriasis. This significance is underscored by the effectiveness of therapies targeting TNF-α [20,21,22]. Produced by various cell types associated with psoriasis, including keratinocytes, neutrophils, dendritic cells (DCs), mast cells, and Th22 cells, Th17, and NKT [23,24], TNF-α exhibits dual effects. On one hand, it notably inhibits plasmacytoid DCs from secreting interferon (IFN)-α [25], often resulting in exacerbated or paradoxical psoriasis during TNF-α inhibitor treatment [26]. Conversely, TNF-α promotes pDCs’ maturation into a more dendritic cell phenotype, facilitating the production of IL-23 [27]. Additionally, TNF-α stimulates the synthesis of IL-18 and IL-12, potent inducers of IFN-γ, thereby contributing to the regulation of the Th1 response [28]. Moreover, TNF-α acts with IL-17A, co-regulating cytokines and keratinocyte genes associated with psoriasis and influencing keratinocyte function [29]. These observations collectively indicate TNF-α‘s pivotal role as a key regulator within the IL-23/IL-17 axis. IL-23 is a heterodimeric protein primarily secreted by DCs, whose levels are elevated in psoriasis [30]. Within psoriatic lesions, the concentration of IL-23 protein notably surpasses that found in unaffected skin [31,32]. Consequently, it can be deduced that IL-23 is intimately linked to the pathogenesis of psoriasis. IL-23 influences T cells through its actions, particularly CD4+ helper T cells (Th17 cells), via a cellular receptor complex [33]. Subsequently, IL-23 stimulates the secretion of IL-17, another crucial cytokine implicated in psoriasis, by Th17 cells through the activation of signaling pathways [34]. The IL-17 family encompasses six structurally akin cytokines, spanning from IL-17A to IL-17F [35]. Investigations have noted IL-17A as the cytokine with the greatest biological activity, exhibiting the highest concentration in psoriasis [36,37,38]. Consequently, IL-17A, also recognized as IL-17, has garnered substantial interest due to its pro-inflammatory characteristics and involvement in autoimmune disorders [39]. The secretion of IL-17, particularly IL-17A and IL-17F, predominantly exerts direct effects on keratinocytes, inducing the synthesis of various molecules such as β-defensins, antimicrobial peptides (AMPs), and cytokines. Additionally, IL-17 stimulates the production of chemokines which are often elevated in psoriatic lesions to recruit neutrophils, lymphocytes, and macrophages [40]. Furthermore, IL-17 has been implicated in promoting keratinocyte proliferation [41]. Psoriasis entails a sophisticated web of interactions among diverse cellular components and molecular entities. At the heart of disease advancement lies the intricate interplay between the adaptive and innate immune systems. This dynamic dialogue triggers the synthesis of numerous cytokines, which uphold characteristic psoriatic manifestations in both the dermis and epidermis. Additionally, keratinocytes contribute to inflammation and local activation. Other pathways implicated in psoriasis, such as CCL20-CCR6, the IL-36/IL-1 pathway, the IFN pathway, and others involving cytokines like IL-22 and IL-6, are under investigation as potential targets for new drug development. A schematic representation of all these interconnections is shown in Figure 1.

### 2.1. TNF-α Inhibitors

The currently approved and available TNF-α inhibitors include etanercept, infliximab, adalimumab, certolizumab, and golimumab [42,43,44]. Etanercept received approval from the US Food and Drug Administration (FDA) in November 1998 as the first anti-tumor necrosis factor agent for the treatment of moderate to severe rheumatoid arthritis (RA) (217). In comparison to etanercept, infliximab has shown a faster and more pronounced response in the early stages of therapy (218). Furthermore, infliximab is effective in resolving skin lesions and reducing joint pain in patients with psoriatic arthritis (PsA) [45]. Adalimumab is an important treatment option for adults with moderate to severe chronic plaque psoriasis and also provides a promising systemic treatment alternative for children and adolescents aged 4 years and older [46]. Certolizumab pegol offers pharmacokinetic benefits due to its lack of an Fc region, which results in minimal placental transfer, low infant exposure during lactation, and reduced oral bioavailability [47]. As a result, certolizumab is a valuable option for the treatment of moderate to severe plaque psoriasis, especially in women of reproductive age. Golimumab, a fully human monoclonal antibody with a bivalent Fab region, shows higher affinity for both soluble and transmembrane forms of the TNF-α protein compared to infliximab and adalimumab, effectively decreasing circulating levels of TNF-α protein and reducing its binding to receptors [48]. Additionally, golimumab has lower immunogenicity than other TNF-α inhibitors. However, treatment with adalimumab has been associated with adverse events such as thrombocytopenia and leukopenia [49]. Table 1 summarizes all recent phase II, III, and IV clinical trials involving anti-TNF-α biologic agents used in the management of psoriasis.

### 2.2. IL-23 Inhibitors

Monoclonal antibodies targeting IL-23 exhibit a direct inhibitory effect on the production of cytokines associated with psoriasis. Currently, IL-23 antagonists such as guselkumab, tildrakizumab, risankizumab, and mirikizumab are prominent in this field. Guselkumab gained FDA approval in 2017 as the first IL-23 inhibitor for treating moderate to severe plaque psoriasis. Its efficacy has been validated across various psoriasis types [50,51]. The VOYAGE 1/2 trials, the initial phase III investigations into guselkumab for psoriasis treatment, involved randomizing patients to receive either guselkumab or adalimumab (administered at 100 mg in weeks 0 and 4 and every 8 weeks). Results indicated significant efficacy with 85% of guselkumab users achieving an Investigator Global Assessment (IGA) score of 0 or 1 by week 16, along with a 73% Psoriasis Area and Severity Index (PASI) 90 response rate, surpassing placebo and adalimumab. Adverse reaction incidence did not significantly differ from other treatments [52]. Tildrakizumab, a humanized monoclonal antibody, has demonstrated sustained efficacy and favorable tolerability in improving plaque psoriatic lesions. It exhibits minimal impact on cardiometabolic risk factors, suggesting a favorable safety profile [53]. Cost-effectiveness analyses position tildrakizumab as one of the most economical first-line treatments for moderate to severe psoriasis [54]. Similarly, risankizumab, an IgG1 monoclonal antibody targeting the p19 subunit of IL-23, exhibits superior therapeutic efficacy compared to ustekinumab and adalimumab [55]. It appears to offer improved efficacy and reduced risk compared to other biologics targeting IL-23 p19, IL-12/IL-23 p40, and IL-17 [56]. In Table 2, all the most recent phase II, III, and IV clinical studies concerning IL-23 inhibitors used for the management of psoriasis are summarized.

### 2.3. IL-17 Inhibitors

The IL-17 cytokine pathway plays a pivotal role in psoriasis pathogenesis, and targeting IL-17 or its receptors with inhibitors represents an effective treatment strategy. Three biologic agents sanctioned by the FDA for psoriasis treatment operate through this pathway, including two monoclonal antibodies with high IL-17A affinity: secukinumab and ixekizumab. Secukinumab, the pioneer monoclonal antibody in this category, received FDA approval in 2015 for psoriasis management [57]. Clinical trials have demonstrated its efficacy in reversing plaque psoriasis histopathology by curbing IL-17A production, thereby facilitating plaque resolution. Notably, pivotal studies like ERASURE and FIXTURE reported impressive PASI 75 response rates, with 77.1% to 81.6% of patients achieving this milestone after receiving 300 mg of secukinumab and 67.0% to 71.6% with 150 mg at week 12 [58]. Beyond plaque psoriasis, secukinumab has shown efficacy in treating nail psoriasis and psoriatic arthritis. Additionally, compared to adalimumab, secukinumab exhibits superior treatment retention in psoriatic arthritis management [59]. Ixekizumab, employing a mechanism akin to secukinumab, exhibits comparable efficacy in psoriasis treatment. Notably, a phase III trial showcased that a minimum of 87.0% of patients with moderate-to-severe plaque psoriasis attained a PASI 75 response with continuous ixekizumab treatment. Additionally, ixekizumab demonstrates superiority over etanercept in clinical efficacy. However, adverse events including infection, headache, neutropenia, and inflammatory bowel disease have been associated with both secukinumab and ixekizumab [60]. Apart from IL-17A inhibitors, brodalumab, targeting IL-17RA directly, has shown efficacy in psoriasis treatment. Phase III trials (AMAGINE-2 and AMAGINE-3) reported PASI 75 response rates of 85.0% and 86.0%, respectively, following biweekly subcutaneous administration of brodalumab. Brodalumab also shows efficacy in treating psoriatic arthritis [61]. Among the IL-17 antibodies approved by the FDA, bimekizumab, a monoclonal IgG1 antibody and the first dual inhibitor of IL-17F and IL-17A, received FDA approval in October 2023. In a multicenter, double-blind, placebo-controlled, randomized phase III trial (BE READY), 91% of patients with plaque psoriasis achieved a PASI 90 response at 16 weeks with bimekizumab, demonstrating quicker and more robust effects than prior IL-17 and IL-23 inhibitors. Additionally, bimekizumab provides significant response rates, with sustained improvement of skin and joint symptoms lasting up to 56 weeks [62]. Recent studies indicate comparable efficacy between bimekizumab and adalimumab for the treatment of psoriatic arthritis and plaque psoriasis, with bimekizumab achieving superior skin clearance compared to secukinumab. However, bimekizumab is associated with an increased risk of oral candidiasis, highlighting the need for further research to balance therapeutic efficacy with safety considerations [63,64]. Table 3 provides a summary of all recent phase II, III, and IV clinical trials involving IL-17 inhibitors for the management of psoriasis.

### 2.4. Additional Interleukin Inhibitors Employed in Psoriasis

Ustekinumab received FDA approval for psoriasis treatment in 2009, marking it as the only IL-12/IL-23p40 inhibitor for this condition [65]. This monoclonal antibody targets the shared p40 subunit of IL-12 and IL-23, thereby blocking the signaling pathways activated by ligand–receptor interactions. Clinical trials PHOENIX 1 and PHOENIX 2 demonstrated 12-week PASI 75 response rates of 66.7% and 67.1% with the 45 mg dose and 66.4% and 75.7% with the 90 mg dose (administered at week 0, week 4, and every 12 weeks thereafter). Ustekinumab also exhibits sustained efficacy, maintaining significant response rates even three years after treatment initiation [66,67]. Comparative studies show that ustekinumab achieves higher response rates than etanercept but lower rates than brodalumab following 12 weeks of standard psoriasis therapy at therapeutic doses [68]. An additional critical pharmacological target in psoriasis treatment is the IL-36/IL-1 axis. This pathway is crucial in the pathogenesis of psoriasis, especially in generalized pustular psoriasis (GPP). Currently, two monoclonal antibodies, imsidolimab and spesolimab, have shown significant efficacy in global clinical trials involving GPP patients [69]. Notably, spesolimab is a novel, selective, humanized antibody targeting IL-36R, representing the first therapeutic agent specifically aimed at the IL-36 pathway for acute GPP treatment. In a phase I trial involving seven subjects, all patients who received a single intravenous dose of spesolimab (10 mg/kg) exhibited rapid lesion improvement within 4 weeks [70]. These findings were supported by the phase II Effisayil-1 trial, which demonstrated that 85% of patients with moderate to severe acute GPP had no visible pustules on their skin after 12 weeks of treatment, and 80% experienced significant remission or complete resolution of skin symptoms [71]. Conversely, the GALLOP phase II trial highlighted the effectiveness of imsidolimab, and this investigation has progressed to phase III based on these promising results [72]. Other anti-IL-1 agents, such as anakinra, gevokizumab, and canakinumab, have also shown promising results in psoriasis treatment [73,74,75]. However, additional large-scale, prospective, randomized clinical trials are needed to confirm the efficacy and safety of these treatments for acute GPP. Furthermore, a significant reduction in IL-8 levels has been observed in refractory psoriatic arthritis patients treated with exogenous IL-36Ra, underscoring the importance of the IL-36 axis in the inflammatory pathways of this condition. Nevertheless, there is a lack of clinical evidence demonstrating the effectiveness of IL-36R-targeted therapies in psoriatic arthritis.

## 3. New and Emerging Oral Small Molecules

In the intricate realm of exploring fresh oral compounds for treating psoriasis, the JAK-STAT pathway emerges as a central player. This signaling route plays a pivotal role in cytokine transmission, fueling inflammation across various autoimmune disorders, thus making it a substantial therapeutic focal point [76]. This pathway orchestrates the activity of numerous cytokines prevalent in psoriasis, including interferon-α, -β, and -γ. Given its pivotal role in psoriasis pathogenesis, there’s a burgeoning interest in exploring novel JAK-STAT inhibitors, complementing those already in use and showcasing promising outcomes, to validate their efficacy in managing this condition [77]. Consequently, multiple ongoing clinical inquiries are meticulously scrutinizing the efficacy and safety profiles of these fresh inhibitors in the realm of psoriasis (Figure 2).

Some medications that have already received approval for psoriasis management and inhibit this pathway are classified as first-generation. For example, tofacitinib, an inhibitor that disrupts the JAK signaling pathway, thereby substantially alleviating all psoriasis-related symptoms, also received full FDA approval following a number of events related to its safety. Prescribability of this drug is possible with appropriate clinical monitoring of the patient. Although tofacitinib outperformed placebo in the treatment of psoriasis, it is not FDA-approved for this specific application due to concerns about its clinical efficacy and long-term safety [78]. Various studies have verified its advantages, and notably, a phase III trial indicated that over 50% of patients on a regimen of 5 mg tofacitinib twice daily and more than 60% of patients on 10 mg twice daily achieved a significant improvement [79]. Another case is upadacitinib, approved in 2022 for psoriatic arthritis, which has demonstrated good results in terms of effectiveness and safety in clinical research. Yet, there are currently no trials planned to assess its effectiveness and safety for psoriasis [80,81]. On the other hand, baricitinib and ruxolitinib target the same pathway and function by inducing Th17 cell apoptosis; both of these are first-generation JAK inhibitors that effectively treat psoriasis [82]. Nevertheless, a careful examination of the safety profile and adverse effects of these drugs is essential, as all first-generation JAK inhibitors target the kinase region of various JAK proteins, heightening the risk of nausea, infections, hemoglobin reduction, and potential gastrointestinal perforation [83]. In Table 4, all the most recent phase II, III, and IV clinical studies concerning JAK inhibitors used for the management of psoriasis are summarized.

Second-generation JAK inhibitors have the capability to selectively target specific JAK proteins without impacting other cytokines; for instance, deucravacitinib (authorized by the FDA in 2022) has targeted effects on TYK2 [84,85]. A recent randomized, double-blind, placebo-controlled phase III trial demonstrated a remarkable therapeutic response after 16 weeks of daily oral administration of deucravacitinib at a dose of 6 mg, when compared to patients who received either placebo [86]. PF-06826647 represents a novel TYK2 inhibitor that recently concluded its phase II trial aimed at assessing its safety and effectiveness in patients with moderate to severe plaque psoriasis [87]. During its initial human study, adverse events and alterations in PASI scores were evaluated in 40 plaque psoriasis patients. These patients were randomly assigned to receive either a placebo or PF-06826647 (400 mg) once daily for a period of 28 days. Notably, patients receiving 100 mg and 400 mg doses of PF-06826647 experienced mean decreases of −14.62 and −24.18 in PASI scores from baseline, respectively, in comparison to −11.13 observed in the placebo group [88,89]. Brepocitinib, a potent TYK2/JAK1 selective inhibitor, is being developed for psoriasis treatment. A phase I trial with 30 patients showed dose-dependent efficacy trends, with higher response rates and greater reductions in PASI scores in the 30 mg and 100 mg brepocitinib groups compared to placebo. Treatment-emergent adverse events were mild, with no serious adverse events reported. Phase II trials further validated brepocitinib’s efficacy, showing significant improvements in PASI scores and response rates compared to placebo across various treatment regimens. Safety data indicated a low risk of treatment-emergent adverse events, primarily mild and treatment-related, with common adverse events including headache, psoriasis exacerbation, and upper respiratory tract infections [90]. While the majority of selective JAK inhibitors are still under scientific investigation and have not yet received market approval, they are viewed as promising therapeutic agents due to their enhanced safety profile. Other potential oral therapies include piclidenoson (CF101), an agonist of the adenosine A3 receptor, which has been found to be overexpressed in inflammatory conditions such as psoriasis or other autoimmune diseases. Preliminary studies have demonstrated its efficacy, but particularly, a very recent study (double-blind phase 3 COMFORT-1 trial randomized-NCT03168256) conducted on 529 patients showed that piclidenoson demonstrated efficacy responses that increased over time alongside a favorable safety profile [91]. These results provide further justification for its ongoing clinical advancement as a treatment for psoriasis. Belumosudil (KD025), a Rho-associated kinase (ROCK2) selective inhibitor, shows efficacy in patients with psoriasis vulgaris. In particular, in an interesting study, some researchers have demonstrated that treatment with belumosulid resulted in a 50% reduction in psoriasis area and severity index PASI scores in 46% of patients. This treatment also led to decreased epidermal thickness and T-cell infiltration in the skin. Additionally, significant reductions in IL-17 and IL-23 levels were observed, along with increased IL-10 levels in responders. These findings indicate that oral administration of belumosulid effectively downregulates the Th17-driven autoimmune response and improves clinical symptoms in psoriatic patients by modulating cytokine levels without adversely affecting the immune system [92]. Vimirogant (VTP-43742) is a potent, selective, and orally active RORγt inhibitor that has shown promising efficacy in phase II trials for patients with plaque psoriasis but has also been linked to certain adverse events [93]. Currently, in another study conducted on patients with moderate to severe plaque psoriasis, another orally active RORγt inverse agonist called Cedirogant (ABBV-157) has shown promising results [94]. Recent scientific studies are also focusing on PDE4, which, having demonstrated powerful anti-inflammatory activity, is drawing the attention of many scientists for the potential involvement of its inhibitors in treating various skin disorders such as psoriasis [95]. In 2014, the oral PDE4 inhibitor apremilast was approved in the United States for adult patients with moderate to severe plaque psoriasis. A clinical study (PALACE 1) demonstrated that apremilast (30 mg twice daily) achieved a response rate double that of the control group given placebo [96]. Another phase III clinical trial (ESTEEM 1) showed the efficacy of apremilast in moderate to severe plaque psoriasis, with even higher response rates compared to placebo [97]. Despite these promising findings, the use of PDE4 inhibitors in psoriasis treatment has been hampered by side effects such as emesis. As researchers continue to explore this therapeutic target, there is hope that ongoing investigations will unlock its full potential and offer new avenues for improving patient outcomes.

Recent research indicates that IL-23, produced by CD301b+ cells, can drive the local proliferation of TRM cells, suggesting that IL-23 inhibitors or new treatments targeting CD301b may provide a promising approach for managing disease relapse and sustaining therapeutic effects. Nonetheless, translating these novel therapeutic strategies into clinical practice requires further investigation. While considerable progress has been made in understanding and treating psoriasis, many aspects remain unexplored. A crucial future objective is to develop effective therapies for managing psoriasis comorbidities, particularly metabolic and cardiovascular disorders [98].

## 4. Conclusions and Prospects

Psoriasis is a chronic inflammatory skin condition that imposes a significant burden on both individuals and healthcare systems, underscoring the urgent need for a comprehensive understanding of the mechanisms driving the disease’s pathogenesis. In the past decade, advances in understanding the immunopathology and pathogenesis of psoriasis have led to significant therapeutic innovations. Targeted biologic therapies have introduced new possibilities for patients, offering substantial disease modification and potentially even curative outcomes by preventing the accumulation of tissue-resident memory T cells in the skin. However, challenges persist in managing psoriasis, including the side effects of medications and the risk of disease relapse following treatment discontinuation. Prolonged use of certain drugs may lead to adverse effects and the development of immune tolerance, thereby limiting their long-term efficacy. Furthermore, the high cost of these therapies often impacts the sustainability of national healthcare systems. A deeper understanding of the mechanisms behind disease recurrence highlights the importance of early intervention to prevent the onset and proliferation of tissue-resident memory T cells. Special attention is also needed for rarer subtypes of psoriasis, such as palmoplantar and pustular psoriasis, which often show limited response to conventional treatments. Additionally, despite the identification of numerous potential metabolic molecules, a thorough understanding of psoriasis’s metabolic effects is still lacking. With the advent of advanced tools like single-cell metabolomics and spatial metabolomics, it will be possible to construct a comprehensive metabolic network of psoriasis through innovative algorithms and integrated analyses. Targeted metabolic therapies are still in their infancy, likely due to metabolic variability, flexibility, and unpredictable side effects. Identifying more effective metabolic targets will be a critical step forward. We anticipate that focusing on psoriasis metabolism and establishing a metabolism-centered psoriasis network will become a prominent research topic in the near future. An important area for future investigation involves evaluating the potential of genetic biomarkers as early indicators for psoriasis, which could facilitate earlier diagnosis and more precise intervention strategies.

## Figures and Tables

**Figure 1 antibodies-13-00076-f001:**
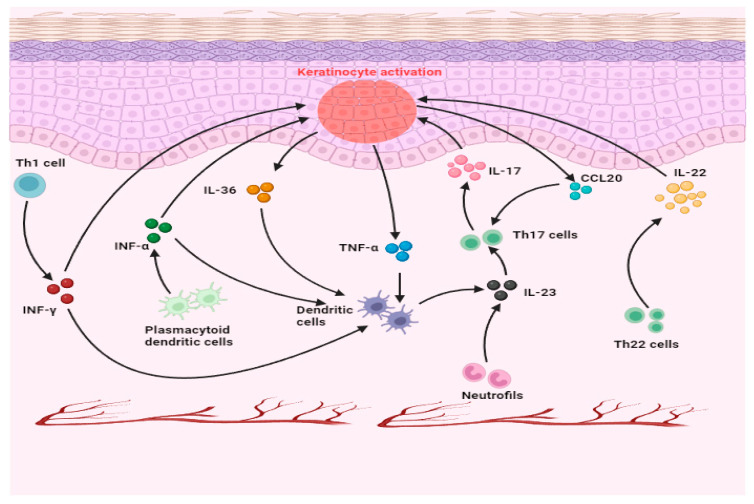
The interplay of cytokine pathways in psoriasis. A sophisticated web connects the fundamental molecules involved in the development of psoriasis. This interconnection is regarded as pivotal for advancement. On one front, the reciprocal enhancement of the adaptive and innate immune systems generates multiple cytokines and sustains characteristic psoriatic traits in both the dermis and epidermis. Conversely, keratinocytes foster the mediators and bolster the proliferation of activation.

**Figure 2 antibodies-13-00076-f002:**
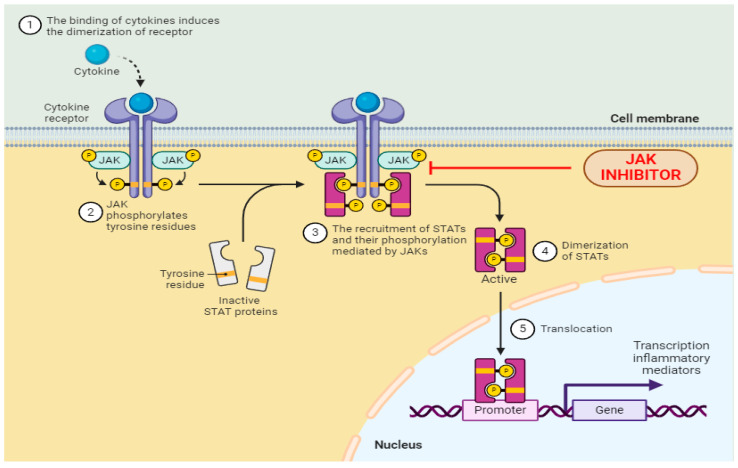
A schematic figure of the JAK/STAT pathway illustrating the mechanism of action of JAK inhibitors. P is the chemical symbol for phosphorus.

**Table 1 antibodies-13-00076-t001:** Biological targeting TNF-α in the management of psoriasis. The table summarizes all treatments and the NCT code to further explore potential therapeutic developments.

Drug Name	Indication	Phase	NCT
Adalimumab	Plaque psoriasis	III	NCT05073315
Adalimumab	Moderate/severe plaque psoriasis	III	NCT02762955
Adalimumab	Plaque psoriasis	III	NCT03316781
Adalimumab	Chronic plaque psoriasis	III	NCT02581345
Adalimumab	Psoriasis, sleep apnea, obstructive	IV	NCT01181570
Adalimumab	Psoriasis, cardiovascular disease	IV	NCT01866592
Adalimumab	Psoriasis, cardiovascular disease	IV	NCT03082729
Certolizumab	Chronic plaque psoriasis, psoriasis	II	NCT00245765
Certolizumab	Chronic plaque psoriasis, psoriasis	II	NCT00329303
Etanercept	Psoriasis	IV	NCT01971346
Infliximab	Psoriatic arthritis	IV	NCT00432406

**Table 2 antibodies-13-00076-t002:** Biological agents targeting IL-23 employed in the management of psoriasis. The table summarizes all treatments and the NCT code to further explore potential therapeutic developments.

Drug Name	Indication	Phase	NCT
Guselkumab	Psoriasis	IV	NCT05858632
Guselkumab	Psoriasis	IV	NCT05004727
Mirikizumab	Plaque psoriasis	II	NCT02899988
Mirikizumab	Psoriasis	III	NCT03482011
Risankizumab	Psoriasis	II	NCT05283135
Risankizumab	Moderate to severe plaque psoriasis	III	NCT03255382
Risankizumab	Psoriasis	IV	NCT04630652
Tildrakizumab	Psoriasis vulgaris	IV	NCT04541329
Tildrakizumab	Psoriasis vulgaris	IV	NCT05390515
Tildrakizumab	Psoriasis	IV	NCT05110313
Tildrakizumab	Psoriasis	IV	NCT04271540

**Table 3 antibodies-13-00076-t003:** Biological agents targeting IL-17 employed in the management of psoriasis. The table summarizes all treatments and the NCT code to further explore potential therapeutic developments.

Drug Name	Indication	Phase	NCT
Bimekizumab	Psoriasis	IV	NCT04340076
Brodalumab	Psoriasis vulgaris	IV	NCT04306315
Brodalumab	Psoriasis vulgaris	IV	NCT03331835
Ixekizumab	Psoriasis (moderate to severe)	II	NCT01107457
Secukinumab	Psoriasis	II	NCT02483234
Secukinumab	Moderate to severe plaque-type psoriasis	III	NCT01365455
Secukinumab	Moderate to severe plaque-type psoriasis	III	NCT01544595

**Table 4 antibodies-13-00076-t004:** Small molecules approved in the management of psoriasis. The table summarizes all treatments and the NCT code to further explore potential therapeutic developments.

Drug Name	Indication	Phase	NCT
Abrocitinib	Plaque psoriasis	II	NCT02201524
Baricitinib	Plaque psoriasis	II	NCT01490632
Peficitinib	Plaque psoriasis	II	NCT01096862
Ruxolitinib	Plaque psoriasis	II	NCT00820950
Ruxolitinib	Plaque psoriasis	II	NCT00778700
Ruxolitinib	Plaque psoriasis	II	NCT00617994
Tofacitinib	Plaque psoriasis	II	NCT01710046
Tofacitinib	Psoriasis, psoriasis vulgaris	II	NCT01831466
Tofacitinib	Plaque psoriasis	III	NCT01815424
Tofacitinib	Plaque psoriasis	III	NCT01309737
Tofacitinib	Psoriasis	III	NCT01163253
Tofacitinib	Psoriasis	III	NCT01519089

## Data Availability

Full availability of data and materials. All stated data can be provided on request to the reader.

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
