# Peer review of "Therapeutic Advances in Psoriasis: From Biologics to Emerging Oral Small Molecules"

_2073-4468, 2024, doi:10.3390/antib13030076_

Round 1
Reviewer 1 Report
Comments and Suggestions for Authors
This is a broad review of the many agents available and coming to treat psoriasis.
1. The following statement is problematic: "For instance, tofacitinib, which has been given the green light by the FDA, is an inhibitor that disrupts the JAK signaling route by binding to the ATP-binding site of the protein, thus substantially alleviating psoriatic arthritis." Tofacitinib received a FDA complete response letter in 2015 in psoriasis. I do not unnderstand how this is a green light: https://www.pfizer.com/news/press-release/press-release-detail/pfizer_receives_complete_response_letter_from_fda_for_oral_xeljanz_tofacitinib_citrate_supplemental_new_drug_application_for_moderate_to_severe_chronic_plaque_psoriasis
The manuscript does provide much information. However, the article provides no information about comparative efficacy or effectiveness of these agents. Readers may come away thinking etanercept and bimekizumab are equally effective. Figures may help the reader to better understand the marked differences between agents.
The work covers emerging oral small molecule agents, but ignores emerginc oral large molecule agents such as: https://pubmed.ncbi.nlm.nih.gov/38324484/
It is unclear to me how much information the tables add with NCT identifiers for each study. What do these tables communicate to the reader?
By discussing emerging agents, the authors choose to not mention existing agents such as methotrexate, cyclosporine and acitretin. This reviewer thinks they are not irrelevant to the present discussion.
Author Response
Comment 1This is a broad review of the many agents available and coming to treat psoriasis. The following statement is problematic: "For instance, tofacitinib, which has been given the green light by the FDA, is an inhibitor that disrupts the JAK signaling route by binding to the ATP-binding site of the protein, thus substantially alleviating psoriatic arthritis." Tofacitinib received a FDA complete response letter in 2015 in psoriasis. I do not unnderstand how this is a green light: https://www.pfizer.com/news/press-release/press-release-detail/pfizer_receives_complete_response_letter_from_fda_for_oral_xeljanz_tofacitinib_citrate_supplemental_new_drug_application_for_moderate_to_severe_chronic_plaque_psoriasis
Accepted, modified and clarified. What this sentence meant was that tofacitinib, despite its many safety problems compared to other drugs (cardiac events), was nevertheless authorised for treatment in psoriasis (also psoriatic arthritis and not just plaque psoriasis) with adequate monitoring of the patient being treated. The sentence was therefore intended to clarify the safety aspect and not that of efficacy, which is undisputed. Reference n.79 highlights this aspect. In red the sentence has been improved to mean this.
The manuscript does provide much information. However, the article provides no information about comparative efficacy or effectiveness of these agents. Readers may come away thinking etanercept and bimekizumab are equally effective. Figures may help the reader to better understand the marked differences between agents.
Accepted and clarified. A sentence in red in the final of introduction is inserted to clarify this. This sentence is also intended to clarify the objectives of this review paper. Of all the drugs proposed and currently on the market, there is no regulatory definition for the use of one drug over another. The patient's clinical condition leads the physician to choose one treatment over another. For example, etanercept may still have some functionality in naïve and selected patients, so in such circumstances, given the lower costs to date with the biosimilar drug of etanercept, this treatment should be preferred as first choice over the newer high-cost treatments for the same clinical benefit.
The work covers emerging oral small molecule agents, but ignores emerginc oral large molecule agents such as: https://pubmed.ncbi.nlm.nih.gov/38324484/
Accepted, clarified and added. Treatments for psoriasis are evolving and expanding. At the previous commentary we are told to give a lot of information, here instead we are highlighted as lacking a new therapeutic perspective. Another reviewer tells us he wants more molecular aspects. Obviously fair comments, but to include everything a book would be more suitable, not an already quite extensive review article. For these reasons we, the authors, decided to do a review only of currently treatments. However to satisfy the commentary we have included in the conclusion a sentence in red that gives this additional perspective with the suggested citation (Ref.98). Thank you for the recommendation which certainly improves the final quality of our work.
It is unclear to me how much information the tables add with NCT identifiers for each study. What do these tables communicate to the reader?
Accepted and clarified. Psoriasis treatments are rapidly evolving, and studies are occurring on various aspects of psoriatic pathology. The reader is provided with a table that is intended to represent the current state of the art of treatments with NCT identifiers for each study so that the reader can learn more about certain aspects of a molecule and see any clinical novelties found by the studies. In addition, the tables bring order and list all the molecules in order to make the review more orderly. “The table summarizes all treatments and the NCT code to further explore potential therapeutic developments” this sentece is been included in the text.
By discussing emerging agents, the authors choose to not mention existing agents such as methotrexate, cyclosporine and acitretin. This reviewer thinks they are not irrelevant to the present discussion.
Accepted and clarified. Even the authors think that “old” oral treatments are not irrelevant, but nevertheless the review has a specific purpose and in line with the focus of the journal. The article goes on to discuss treatments for “moderate to severe” psoriasis and in particular focuses on monoclonal antibodies and extends to cover small molecules recently introduced to the market. The title says this, “Therapeutic advances in psoriasis: From biologics to emerging oral small molecules.” That is, talk about late-stage treatments starting with biologic drugs. Earlier treatments that go beyond the scope of the article are consequently not discussed although they may still have residual importance. This is included in red in the objectives at the end of the introduction to clarify, but we think it is already intuitive enough.
We thank you for all suggestions and hope to have satisfied all comments in order to receive final acceptance.

Reviewer 2 Report
Comments and Suggestions for Authors
This review summarized the therapeutic drugs for the treatment of psoriasis. In general, this review is well written. As the authors mentioned, the metabolic therapy is still in its early stages, however, the metabolic pathways, especially glucose and lipid metabolism were greatly changed in psoriatic samples. For example, glucose is required for the fast proliferation of keratinocytes in psoriatic skin. The dysregulated lipid metabolism may further disrupt the skin barrier and exacerbate skin inflammation. All these points can be discussed in the review. Except for drugs related to immunity regulation and oral medications, topical medications are also promising in treatment of psoriasis, as these medications have less side effects owing to the local efficacy. At last, the combination treatment with different drugs is also worth to be mentioned.
Author Response
This review summarized the therapeutic drugs for the treatment of psoriasis. In general, this review is well written. As the authors mentioned, the metabolic therapy is still in its early stages, however, the metabolic pathways, especially glucose and lipid metabolism were greatly changed in psoriatic samples. For example, glucose is required for the fast proliferation of keratinocytes in psoriatic skin. The dysregulated lipid metabolism may further disrupt the skin barrier and exacerbate skin inflammation. All these points can be discussed in the review. Except for drugs related to immunity regulation and oral medications, topical medications are also promising in treatment of psoriasis, as these medications have less side effects owing to the local efficacy. At last, the combination treatment with different drugs is also worth to be mentioned.
Thank you for your positive comment on our work. As it is clearly represented, this review focuses on psoriasis treatments, analysing diagnosis and molecular macanisms in a summary and introductory way. Per questi motivi la discussione riguardanti le vie metaboliche ha un discorso marginale rispetto all'argomento principale che è quello riguardante i trattamenti farmacologici. It would certainly have been interesting to go deeper into molecular and pathological discourses, but this review is already quite extensive and we plan to do a further review that deals more with metabolic phenomena with the help of more competent authors. With regard to topical treatments these are not included in the ‘moderate to severe plaque psoriasis therapy’, topical treatments have a local symptomatic role which is outside the scope of the review. Furthermore, the article focuses on monoclonal antibodies to fully enter the scope of the paper and to date no systemic combination therapies are recommended. We hope we have satisfied all comments and curiosities. Some possible changes were made, however, based also on suggestions from other reviewers.
We thank you for all suggestions and hope to have satisfied all comments in order to receive final acceptance.

Round 2
Reviewer 1 Report
Comments and Suggestions for Authors
This manuscript is improved.
However, because the manuscript not only discusses approved medications but highlights drugs in development, there was the absence of mention of the oral IL-23 agent (JNJ-2113). This is an emerging approach to treating chronic inflammatory skin diseases and belongs in this discussion.
Author Response
This manuscript is improved.
However, because the manuscript not only discusses approved medications but highlights drugs in development, there was the absence of mention of the oral IL-23 agent (JNJ-2113). This is an emerging approach to treating chronic inflammatory skin diseases and belongs in this discussion.
Accepted and added.
Also highlighted in red in the new oral molecules section is the opportunity of IL-23 with NEJM citation 98.
Thank you for help in the review to make a complete picture of current and future treatments.
